# Size Matters: The Functional Role of the CEACAM1 Isoform Signature and Its Impact for NK Cell-Mediated Killing in Melanoma

**DOI:** 10.3390/cancers11030356

**Published:** 2019-03-13

**Authors:** Iris Helfrich, Bernhard B. Singer

**Affiliations:** 1Skin Cancer Unit of the Dermatology Department, Medical Faculty, University Duisburg-Essen, West German Cancer Center, Essen 45147, Germany; 2German Cancer Consortium (DKTK) partner site Düsseldorf/Essen, Essen 45147, Germany; 3Institute of Anatomy, Medical Faculty, University Duisburg-Essen, Essen 45147, Germany; bernhard.singer@uk-essen.de

**Keywords:** CEACAM1, NKG2D ligands, melanoma, NK cells

## Abstract

Malignant melanoma is the most aggressive and treatment resistant type of skin cancer. It is characterized by continuously rising incidence and high mortality rate due to its high metastatic potential. Various types of cell adhesion molecules have been implicated in tumor progression in melanoma. One of these, the carcinoembryonic antigen-related cell adhesion molecule 1 (CEACAM1), is a multi-functional receptor protein potentially expressed in epithelia, endothelia, and leukocytes. CEACAM1 often appears in four isoforms differing in the length of their extracellular and intracellular domains. Both the CEACAM1 expression in general, and the ratio of the expressed CEACAM1 splice variants appear very dynamic. They depend on both the cell activation stage and the cell growth phase. Interestingly, normal melanocytes are negative for CEACAM1, while melanomas often show high expression. As a cell–cell communication molecule, CEACAM1 mediates the direct interaction between tumor and immune cells. In the tumor cell this interaction leads to functional inhibitions, and indirectly to decreased cancer cell immunogenicity by down-regulation of ligands of the NKG2D receptor. On natural killer (NK) cells it inhibits NKG2D-mediated cytolysis and signaling. This review focuses on novel mechanistic insights into CEACAM1 isoforms for NK cell-mediated immune escape mechanisms in melanoma, and their clinical relevance in patients suffering from malignant melanoma.

## 1. Immunosurveillance and NK Cell Signaling in Melanoma

Melanomas often originate from benign nevi consisting of clonally expanded, highly proliferative melanocytes without the tendency for progression. The abnormal proliferation of melanocytes can lead to the development of dysplastic nevi, potential precursor lesions of melanoma, which are characterized by a low invasive potential. Early melanomas have the potential to metastasize to distant organs via induction of the vertical growth phase. Consequently, melanoma is the most aggressive type of skin cancer with cutaneous melanoma the cause of about 55,500 cancer deaths annually worldwide [1]. Tumor progression starts quite early and metastatic seeding to distant sites causes a median survival of 6 to 12 months in patients with advanced melanoma [2]. Both genetic and environmental factors are involved in the development of melanoma, with excessive exposure to Ultraviolet (UV) radiation being the most important risk factor. Indeed, a positive correlation between childhood sunburn and subsequent risk of cutaneous melanoma is now well established [3,4]. Furthermore, the malignant transformation of melanocytes is caused by the reciprocal interplay between exogenous and endogenous triggers as well as tumor-intrinsic and immune-related factors. Consistent with other cancers, the constant increase of point mutations and gene copy number alterations drives malignant transformation and constitutive activation of the oncogenic mitogen-activated protein kinase (MAPK) signaling pathway [5]. As such, tumors represent multi-cellular organs composed by a full flower of different cell types including a broad range of immune cell types and stromal cells, like activated fibroblasts and endothelial cells. The multicellular interplay coordinates a continuum between tumor elimination, equilibrium, and escape, depending on the state of the immune system and molecular properties of the cancer cells. Due to the high immunogenicity of melanoma, presented by its historical response to interferon alpha (IFN-α) [6] and interleukin 2 (Il-2) [7], melanoma cells are able to induce a cytotoxic T cell-mediated immune response [8,9].

The development of various immunotherapeutic strategies, especially with regard to so called ‘immune checkpoints’, targeting the cytotoxic-T-lymphocyte-associated protein-4 (CTLA-4) or the programmed cell death 1 (PD1) receptor and its ligand PD-L1, have revolutionized the field of melanoma therapy in recent years [10,11]. Melanoma can evade the immune system through the expression of PD-L1, which binds to PD1 present on activated lymphocytes. PD-L1 interaction with PD1 causes immune tolerance through apoptosis of the activated lymphocytes. In consequence, high level of PD-L1 expression presented on the membrane of tumor cells has been correlated with poor prognosis in melanoma patients [12]. Regarding therapeutic intervention by using checkpoint inhibitors, high PD-L1 expression on melanoma cells has been correlated with favorable clinical outcomes [1,11]. Nevertheless, recent data indicate that melanoma patients who lack expression of PD-L1 on melanoma cells can also respond to PD-1/PD-L1 inhibitors, although to a lower level [13,14]. Apart from the cytotoxic T cell-mediated immune response and corresponding promising therapeutic approaches, the down-regulation of the human leukocyte antigen (HLA) class I molecules on melanoma cells represents a considerable problem for T cell-based immunotherapy [15]. In this context, natural killer (NK) cells, which are a lineage of innate lymphocytes, have gained increasing attention as anti-melanoma effector cells due to their preferential targeting of tumor cells with low HLA class I expression [16,17]. HLA class I molecules are the ligands for major inhibitory NK cell receptors and typically appear in normal cells, thus ensuring tolerance. The balance of stimulatory and inhibitory signals of germ line encoded receptors regulates the activation of NK cells. The ligands for these activating receptors are absent on healthy cells, so tolerance towards a ‘healthy self’ is guaranteed [18,19]. Since their discovery in the 1970s, many studies have identified the major role of NK cells as a first line of defense against transformed tumor cells or virus-infected cells. Moreover, NK cells have the potential to impact the innate immune response through their capacity to release high amounts of cytokines and chemokines. In particular, the CC-chemokine ligand 5, XC-chemokine ligand 1 (XCL1) and XCL2, which results in enhanced recruitment of dendritic cells (DCs) into solid tumors and subsequent improved survival in cancer patients [20]. The most important activating NK cell receptors are; NK gene complex group 2 member D (NKG2D), that binds to molecules of the major histocompatibility complex (MHC) class I related Chain A/B (MICA/B) and UL16 Binding Proteins (ULBP) 1 to 6 on tumor cells; DNAM-1, via binding to CD112 (Nectin-2) and CD155 (also known as poliovirus receptor); and the family members of natural cytotoxicity receptors (NCRs) such as NKp30, NKp44 and NKp46, first identified by Moretta et al. [21]. It has already been reported that NK cells target melanoma cells for lysis via interaction of NKG2D, Nkp46 and DNAM-1, also in the presence of MHC class I molecules [22,23]. Interestingly, soluble ULBP2 has been identified as an independent predictor of prognosis in melanoma patients [24]. Furthermore, NK cell accumulation in tumor-infiltrated lymph nodes of melanoma patients results in NK cell-mediated killing via the release of perforin and granzyme. Perforin and granzyme enter the target cell in contact with the NK cell and induce apoptosis. Additionally, chemokines such as CCL3/MIP-1α and CCL4/MIP-1β, and pro-inflammatory cytokines such as TNF-α, IFN-γ, IL-5, IL-10 and IL-13 are secreted in the local microenvironment and mediate various effects including recruitment of inflammatory cells to the tumor and in some cases even cell death in surrounding tumor cells [18,25,26]. Additionally, the detection of a specific CD56^bright^CD16^+^NK cell subset in metastatic lymph nodes which generally include CD56^bright^CD16^dim/neg^ poorly cytotoxic NK cells, suggested that nodal NK cells may represent an attractive therapeutic target for advanced melanoma patients [27,28]. Interestingly, recent studies indicate a coherency of NK cell mediated-cytotoxicity with the BRAF^V600E^ mutation status, which is known to be presented in nearly 50% of melanoma lesions [29]. It has been described that NK ligand expression depends strongly on the signaling route activated by the oncogenic *BRAF* gene. In consequence, recent in vitro data has shown that, under pressure of the BRAF inhibitor Vemurafenib (PLX4032), human melanoma cells downregulate B7-H6, MICA, ULBP2 and the DNAM-1 ligand CD155, and upregulate MHC class I expression, in order to escape NK-cell mediated tumor cell recognition [30,31].

## 2. CEACAM1 Signaling and Its Function in Melanoma

Uncontrolled proliferation, derangement of cellular and morphological differentiation, invasion and metastatic spread are hallmarks of malignant transformation. Such characteristics can at least partially be attributed to alterations in adhesion and cell–cell communication between neoplastic and normal cells. Thus, melanoma cells escape control from their neighboring keratinocytes and other cell types in their surrounding microenvironment through down-regulation of cell–cell and cell–matrix adhesion molecules, as well as cell–cell communication receptors. The adhesive functions of cell adhesion molecules in homophilic and heterophilic interactions differ with respect to their quality. While integrins and cadherins mediate high affine adhesion, and thus can act as glue between cells and between cell and matrix, members of the immunoglobulin superfamily cell adhesion molecules (IgCAMs) facilitate significantly less affine cell–cell interactions, so mediate touching between cells rather than glue like interactions. Malignant transformation is often accompanied by down-regulation of cell adhesion molecules, which explains, at least partially, the diminished involvement of malignant cells in the tissue association. Melanoma progression is a complex multistep process orchestrated by a variety of cellular factors, including the dysregulation of cell adhesion molecules [32]. Evidence has amassed that the multi-functional carcinoembryonic antigen (CEA)-related cell adhesion molecule 1 (CEACAM1), also known as CD66a, BGP, C-CAM, is a major player in the process of malignant progression.

CEACAM1 belongs to the CEA family within the immunoglobulin superfamily [33] and can be expressed in human epithelial [34,35], endothelial [36], and hematopoietic cells [37,38]. It is heavily N-glycosylated with more than 60% of the mass contributed by glycans, which positively influence the protein stability and half-life. As with most IgCAMs, it mediates low affine cellular interactions with neighboring cells and soluble CEACAM variants in a homophilic fashion. In addition, it can also bind in a heterophilic manner to other members of the CEA family, namely CEACAM5, CEACAM6, and CEACAM8 [39,40]. These interactions profoundly influence a variety of signaling events, including those involved in mitogenesis, survival/apoptosis, differentiation, migration, invasion, the arrangement of three-dimensional tissue structure, angiogenesis, tumor suppression, and the modulation of innate and adaptive immune responses [41,42]. In humans, CEACAM1 is characterized by numerous isoforms generated by alternative splicing mechanisms of exon 5 (A2 domain) and 7 (cytoplasmic domain) [43]. All CEACAM1 variants share one membrane distal IgV-like domain (N-domain) modulating homophilic or heterophilic interactions, and two or three IgC-like domains for a total of 3 (CEACAM1-3) or 4 (CEACAM1-4) heavily glycosylated extracellular domains. These isoforms transmembrane anchored and linked to either a short (S) or a long (L) cytoplasmic domain consisting of 10 or 73 amino acids, respectively [44]. The CEACAM1-L variants contain two immunoreceptor tyrosine-based inhibitory motifs (ITIMs) that serve as a target for various tyrosine kinases and as docking sites for SH2 domains of certain phosphatases like the SHP-1 and SHP-2 tyrosine phosphatases and the Src homology 2 (SH2) domain containing inositol polyphosphate 5-phosphatase (SHIP) (Figure 1). Phosphorylation of CEACAM1 was associated with its effect on cell proliferation and for maintaining contact inhibition [45,46]. In epithelial cells CEACAM1-L was found on both the apical and the lateral surfaces, whereas CEACAM1-S appeared exclusively on the apical surfaces, further supporting the idea of the different functional potential of CEACAM1-L and -S forms and their dimeric complexes in cis and trans [47,48]. In addition, the Ser503 residue of CEACAM1-L seems to be crucial for the inhibition of cancer cell tumorigenicity [49]. It is important to note that, CEACAM1-L, but not CEACAM1-S, negatively regulates proliferation via its ITIM domain [44]. It also binds paxillin if tyrosine phosphorylated and contains a talin and β-catenin binding site in its cytoplasmic tail [50,51]. A common feature of CEACAM1-L and -S is that they interact with the actin cytoskeleton and further proteins like, tropomyosin, filamin A, and calmodulin [36,45,47,50,52,53,54,55]. The CEACAM1-filamin A interaction has been shown to regulate cell migration and to be critical for the formation of an efficient immunological synapse in T-cells [56,57]. These findings fit well with reports showing that CEACAM1 cooperates with the T-cell immunoglobulin domain and mucin domain-3 (TIM-3, also known as HAVCR2). TIM-3 was described as an activation-induced inhibitory molecule, which under certain conditions also has a stimulatory action on T cells. Thus, TIM-3 seems to share the co-stimulatory and co-inhibitory potential with CEACAM1. Although a biochemical heterodimer of TIM-3 and CEACAM1 was stated when first reported, it has since been accepted that the molecular co-localization of these proteins occurs due to the formation of an immunological synapse in T-cells [56,58,59]. However, the exact mechanisms of when and how CEACAM1 and TIM-3 modulate T-cell activation and inhibition remains to be clarified. All these findings underline the different functional potential of CEACAM1-L and -S. What makes the CEACAM1 driven signaling even more complex, is the fact that CEACAM1 dimerizes not only in ‘trans’ but also in ‘cis’, giving rise to L/L, L/S and S/S constellations, which seem to cause different functional effects [48,53]. Öbrinks group very nicely showed that the trans-homophilic binding mediated by the N-domain induces cis-dimerization of CEACAM1. The cis-homodimerization of CEACAM1-L brings its cytoplasmic domains together, thus changing the relative binding affinities for SHP-1/2 and c-Src. Consequently, increased CEACAM1-L homodimerization results in increased binding and activation of SHP-1/2. CEACAM1-S and can interfere with the CEACAM1-L homodimer by forming a heterodimer with CEACAM1-L, thus decreasing the level of the CEACAM1-L cytoplasmic dimer configuration. Consequently, the CEACAM1 driven signaling mechanisms and functional outcome can be modulated by variations of the CEACAM1-S/CEACAM1-L expression ratios [45,60]. 

Although in recent years the potential value of CEACAM1 as a clinically highly relevant diagnostic and therapeutic target for various malignant diseases has gained increasing attention, its role in different cancer entities appeared rather complex. CEACAM1 has been described as a tumor suppressor and inhibitor of proliferation in several tumor entities, with the exception of melanoma [42,45,60,61,62]. However, it was also reported as a driver of invasion [61,63]. In colon carcinomas, hepatocellular carcinomas, a proportion of breast cancers, prostate cancer, and bladder cancer, CEACAM1 is down-regulated supporting the assumption that CEACAM1 acts in epithelial cells as a growth/tumor suppressor [64,65,66]. On the other hand, CEACAM1 appeared overexpressed or neo-expressed in thyroid cancer, gastric cancer, and malignant melanoma [61,63]. Here, its expression is associated with unfavorable prognosis and development of metastasis [41,42,63,67]. CEACAM1 was shown to interact with the beta 3 integrin subunit via the cytoplasmic tail of CEACAM1. By interaction with β3 integrin, CEACAM1-L enhanced the migratory and invasive potential of melanoma cells [61]. In line with this finding, it has been shown that the expression of CEACAM1 in primary cutaneous melanoma predicts the development of metastatic disease [63]. In addition, soluble CEACAM1 level in sera from melanoma patients inversely correlates with overall survival [68,69]. This has spurred discussion of CEACAM1 as a more specific and sensitive biomarker than the currently used Melan-A, S100β and HMB45, and has implicated CEACAM1 as a potential novel therapeutic target in melanoma [70]. One reason for the contradictory expression pattern and functional role of CEACAM1 could be caused by alterations of the CEACAM1-S/CEACAM1-L ratios, which, for example, has been shown to be different in non-proliferating versus proliferating epithelial cells [45,60]. In addition, alterations in the CEACAM1-L/S ratio were already shown to promote the growth and metastasis of non-small cell lung carcinoma (NSCLC) [71]. Analyses of a large cohort of melanoma cell lines revealed a CEACAM1 expression rate of 72% [72]. All CEACAM1-positive cell lines expressed the CEACAM1-4L variant alone or in combination with other CEACAM1 variants. In addition, expression levels of the L-variants (both CEACAM1-3L and CEACAM1-4L) dominated against the S-variants. Interestingly, overexpression of CEACAM1-4L in originally CEACAM1 negative melanocytes and melanoma cell lines increased the migratory and invasive growth potentials, at least in vitro, supporting the role of CEACAM1-4L in melanoma progression and metastasis. One functional principle of CEACAM1 is that it barely does anything alone but influences many other molecules in their function. Supporting this idea, CEACAM1 was shown to inhibit cell-matrix adhesion, thus promoting cell migration, by regulating the expression of N-cadherin [73]. Analysis of tissue biopsies from melanoma patients that, according to the American Joint Committee on Cancer, spanned stages I to IV, revealed also neo-expression of CEACAM1 in around 80% of biopsies [68,72,74]. Interestingly, early in tumor establishment (stage I/II), CEACAM1-4L and CEACAM1-4S were solely expressed, whereas CEACAM1-3L was predominantly and CEACAM1-3S exclusively expressed in progressed melanoma (III/IV). Thus, the lack of CEACAM1-3S expression in early melanoma (stage I/II) followed by the dramatic increase in de novo expression of 86% in higher malignant stages (III/IV) highlights the underestimated role of CEACAM1 variants on individual biological cellular functions, and identified CEACAM1-3 isoforms, in particular the CEACAM1-3S, as potential biomarker to monitor melanoma progression [72].

Most often the leading course of cancer-related deaths arises from widespread metastasis and not the primary tumor itself. This is caused by the high rate of resistance to current therapies, triggered by genetic alterations as well as tumor cell heterogeneity of metastases. The process of metastatic dissemination is directed by multiple cascades. The first step occurs in the primary tumor where subpopulations of tumor cells lose their cell–cell contact via cleavage of the extracellular matrix by matrix metalloproteases. This enables them to exit the tumor mass and invade locally. Following intra-and extravasation of the vascular system, tumor cells need to have the potential to adapt and survive in the foreign environment of distant organs, where they can then form micro- and finally macro-metastases [75,76]. Significantly, CEACAM1 was also shown to control the epithelial-mesenchymal transition by site-specific regulation of beta-catenin [77]. Therefore, extensive efforts have been made to understand the molecular and cellular mechanisms driving the metastatic cascade with the aim of developing new therapeutic approaches against disseminated cancer cells. Data from 3D colony forming assays addressing the individual impact of CEACAM1 isoforms on functional behavior of melanoma cells has identified the CEACAM1-4L version for its potential to drive anchorage-independent growth. In contrast, expression of induced CEACAM1-4S resulted in damped colony formation [78]. Thus, an understanding of the pathways that give rise to tightly-controlled expression of CEACAM1 isoforms may offer new insights into the process of metastatic spread in melanoma. In this regard, the microphthalmia-associated transcription factor (MITF), a key regulator of melanoma proliferation and invasiveness that orchestrates key developmental and differentiation programs in the melanocyte lineage, has recently been identified as a potential regulator of CEACAM1 by a two-step DNA microarray strategy [79]. Analysis of “The Cancer Genome Atlas” (TCGA) melanoma dataset confirmed a positive correlation of MITF-CEACAM1 axis [80]. MITF is the master regulator of the primary pigmentation enzymes Tyrosinase (TYR), TYR-protein 1 (TYPRP1) and TYRP2, which are required for melanin synthesis [81]. It also controls expression of the PMEL17 (gp100) and MLANA (MART-1); both factors required for the formation and maturation of melanosomes [82,83]. MITF controls the cell cycle and survival of melanocytes by regulating transcription of various target genes, including cyclin-dependent kinase 2 (CDK2), T-box transcription factor 2 (TBX2), CDK inhibitors p21^CIP1^ and p16^INK4A^ [84]. Mass spectroscopy and tandem affinity purification identified interaction of MITF with proteins directing DNA damage repair and replication, as well as components of the Polybromo and Brahma-related gene 1 (BRG1)-associated factor (PBAF) chromatin remodeling complex, in melanoma cells [85]. Chromatin-IP followed by high-throughput sequencing (Chip-seq) identified an M-box motif (TCATGTG) representing a potential binding site for MITF located within the CEACAM1 promotor [80]. Additionally, further molecules like the B- and T-cell receptor, the Toll-like receptor 2 and 4, the EGFR, the VEGFR-1 -2 and -3 and the G-CSF receptor, interact with and are functionally modulated through CEACAM1 [56,86,87,88,89,90,91,92]. Furthermore, CEACAM1 has been shown to be involved in intercellular communication processes of various leukocyte subtypes with each other and with CEACAM1 positive epithelial and endothelial cells. Subsequent signal transduction cascades modulated by CEACAM1 often leads to altered functional outcomes [92]. In B-lymphocytes, CEACAM1 is a crucial regulator of survival, where it co-stimulates the B-cell receptor and consequently B-cell numbers, thus improving the protective antiviral antibody response [56,93]. In monocytes CEACAM1 regulates phosphatidylinositol 3-kinase (PI3K) and Akt-dependent survival signal and inhibits apoptosis [94]. A similar anti-apoptotic effect was also shown in granulocytes [38]. In addition, CEACAM1 was demonstrated to initiate the binding to endothelium via E-selectin [95]. The anti-apoptotic effect was also found in T-lymphocytes e.g., preventing CD8+ T cell exhaustion and triggering enhanced T cell proliferation [88]. In CD4+ T-cells, CEACAM1 was shown to augment the IL-2 production and STAT5 phosphorylation leading to enhanced Treg induction and stability, which led to protection from T-cell-mediated liver injury [96]. Although these latest reports clearly documented that CEACAM1 has a central meaning in mediating T- and B-cell driven immunological responses by anti-apoptotic and TCR/BCR co-stimulatory actions, others have stated an inhibitory, anti-inflammatory CEACAM1 effect [44]. Kammerer et al. (1998) nicely pointed out this contradiction, by showing both a co-stimulatory and a co-inhibitory T cell receptor driven immunological reaction of CEACAM1 [87]. Whether this contradiction is caused by different ratios of CEACAM1-L to -S, by a diverse activation or differentiation status of the T-cells corresponding with altered intracellular alterations in the intracellular response machinery, or by other receptors (e.g., TLRs), must be further investigated.

Interestingly, CEACAM1 and other CEACAMs also serve as specific receptors for a number of commensal pathogens like Neisseria gonococci, Moraxella catarrhalis and Helicobacter pylori [97,98,99]. These pathogens have evolved their own proteins, namely Opa (*Neisseria*), UspA1 (*Moraxella*) and HopQ (*H. pylori*) to specifically interact with the identical section in the N-domain of human CEACAM1 and other human CEACAMs. In contrast to the low affinity CEACAM–CEACAM interaction, the binding of Opa/UspA1/HopQ proteins to CEACAMs is mediated by a high affinity binding, leading to the assumption that once it has bound, it will not let go. Thus, CEACAM1 mediates bacterial adherence leading to transcellular transcytosis, suppression of immune cell activities, reduced inflammatory responses, and thus allows the commensal colonization of tissues [92,100]. Unfortunately, the inhibition of pro-inflammatory actions caused by pathogens binding to CEACAM1 may have severe side effects, like triggering acute exacerbation of chronic obstructive pulmonary disease (AECOPD) and maybe even support the immune escape of cancer cells.

## 3. Regulation of NK Cell Function by CEACAM1

Tumors have developed several mechanisms to escape from immune surveillance. On the tumor cell surface, CEACAM1 has been shown to interact directly with CEACAM1 on NK cells and tumor-infiltrating lymphocytes (TIL), leading to functional inhibition of those cells [101,102,103,104]. Following activation, it is mainly the CD16-negative NK subset that expresses CEACAM1 and cells expressing the CEACAM1 are protected from lysis by CEACAM1-positive NK cells [101,105]. It is important to note that, while freshly isolated T- and B-cells express very low amounts of CEACAM1, freshly isolated NK cells were CEACAM1 negative [106]. However, the CEACAM1 expression on NK cells (and T cells) depends on their differentiation stage and can be strongly enhanced by activation with interleukin-2 (IL-2), IL-12 and IL-18, which may lead to distinct regulatory mechanisms of the functional subpopulations of NK cells (Figure 2) [107,108]. The central meaning of the homophilic CEACAM1 interaction in trans was also supported by reports showing that CEACAM1 positive NK cells inhibited cytolysis of CEACAM1 expressing tumor cells by impairing NKG2Ds ability to stimulate cytolysis. Additionally, evidence has amassed that CEACAM1 is involved in the regulation of the NKG2D-ligand expression [72,101,109]. Indeed, CEACAM1 can down-regulate MICA/B and ULBP1 from the tumor cell surface thus influencing their immunogenicity [110,111]. Several studies have shown that soluble NKG2D-ligands derived from cancer cells by alternative splicing [19], proteolytic shedding [20,21,22,23], or exosome secretion [24] can impair NKG2D-mediated cytotoxicity by negatively regulating NKG2D surface expression.

The co-engagement of NKG2D and CEACAM1 resulted in their molecular association and the subsequent recruitment of Src homology phosphatase 1 by CEACAM1-L led to the inhibition of the downstream signaling critical for the initiation of cytolysis [112]. CEACAM1 expressing tumor cells caused intracellular retention of various NKG2D ligands. In contrast, CEACAM1-negative tumor cells expressed more cell surface NKG2D ligands and exhibited greater sensitivity to NK cell mediated cytolysis [112,113]. Importantly, NKG2D is an activating NK cell receptor expressed on all NK cells and CD8+ T cells, as well as a set of CD4+ T cells [114]. NKG2D plays a central role in NK cell and T cell mediated immunity against cancers, infections, and autoimmune diseases [115]. NKG2D exerts its effects through recognition of its cognate ligands on target cells, resulting in their cytolytic destruction. The CEACAM1 based NK cell mediated inhibition of melanoma cell cytolysis was shown to act independently of the MHC class I recognition [101]. Thus, CEACAM1 was attributed to be a novel inhibitory receptor on activated NK cells with adverse effects for tumor immunity. Furthermore, the heterophilic interactions of CEACAM5 (often better known as CEA) with CEACAM1 also inhibited the NK cell mediated killing of tumor cells [104]. Although these findings offered a novel role for the CEACAM5/CEA protein, enabling the escape of tumor cells from NK-mediated killing, CEA is expressed on and released by a wide range of carcinomas, but not melanoma cells. In addition, the CEACAM1 interactions seemed also to be important in some cases of metastatic melanoma, as increased CEACAM1 expression was observed on NK cells derived from such patients if compared with NK cells from healthy donors [101]. A further role for CEACAM1 on NK cells was demonstrated in patients with low level HLA class I molecules (TAP2-deficient patients) and in women during pregnancy [102,116]. These reports demonstrated that CEACAM1 prevents auto-reactivity of NK cells in TAP2-deficient patients and in conditions requiring a strict control of immune effector functions, e.g., during pregnancy at the maternal–fetal interface.

Nevertheless, the detailed function of the four individual CEACAM1 variants in the context of NK-cell mediated antitumor immunity in melanoma is still not fully understood. However, we and others have already showed that CEACAM1 variants differing in their extracellular and intracellular lengths, impact melanoma progression and immune-surveillance in a variant-specific mode of action [61,72,78,113]. Moreover, the functional effects triggered by these distinct CEACAM1 splice variants seem to act differently in different tumor identities. While Chen et al. described down-regulation of NKG2D ligands via expression of CEACAM1-3S and -3L in colon cancer cells [113]. CEACAM1 variants appear to act differently in melanoma. Melanoma patients benefit from the expression of CEACAM1-3S as a consequence of enhanced expression of the NKG2D ligands MICA, ULBP2 and DNAM-1 ligand CD155 on the surface of melanoma cells, thus presenting these cells as targets for NK cell mediated cytolysis [72]. Interestingly, CEACAM1-4L seems to act in an antagonistic mode of action. Expression of CEACAM1-4L on the surface of melanoma cells reduces the expression of MICA, ULBP2, CD155 and CD112 via matrix metalloproteinase (MMP)-mediated shedding of these ligands, resulting in escape of NK cell mediated killing [72].

## 4. Conclusions

Different cell adhesion molecules belonging to the integrin, cadherin and immunoglobulin super families have been implicated in melanoma progression. One of these adhesion molecules, named CEACAM1, is completely absent in normal melanocytes but highly expressed in most melanoma cells. Thus, many studies have linked CEACAM1 expression with melanoma progression and metastasis. Consequently multiple preclinical studies, but also clinical data in patients suffering from different cancer diseases, point to the significance of CEACAM1 as a novel therapeutic target. Moreover, its potential to enhance efficacy to cancer immunotherapy is due to its well-established function in T and NK cells. The first clinical trials using CEACAM1-specific antibodies have been initiated, and the results of these trials will hopefully shed light on the therapeutic mode of action.

## Figures and Tables

**Figure 1 cancers-11-00356-f001:**
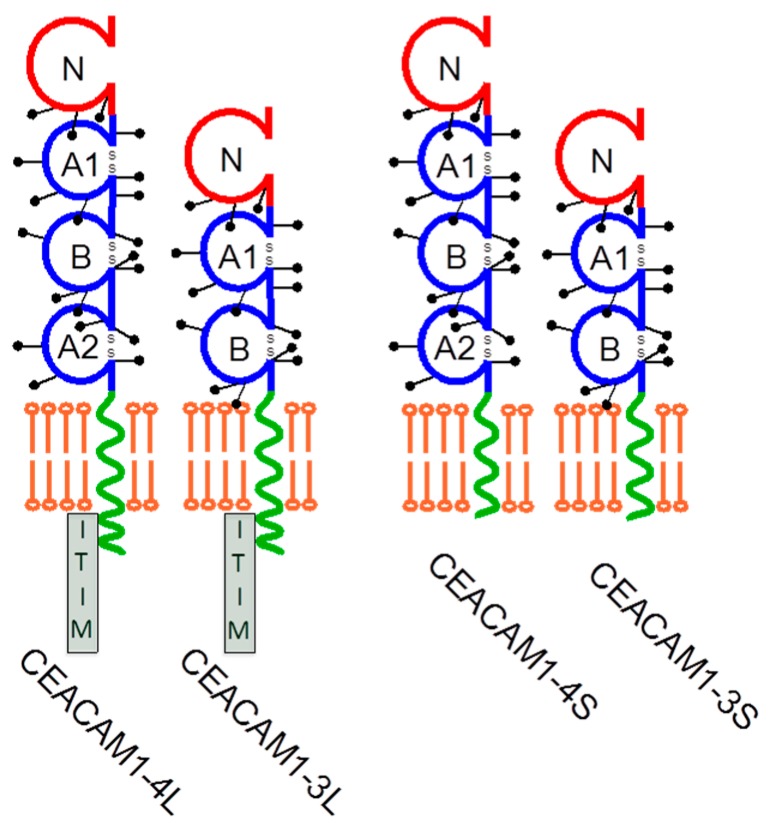
The four mainly expressed splice variants of human CEACAM1. Alternative splicing of CEACAM1 generates isoforms, which consists of one N-terminal ectodomain followed by either three or four extracellular Ig-like domains, a conserved transmembrane domain, and a short (CEACAM1-4S and 1-3S) or long cytoplasmic (CEACAM1-4L and 1-3L) domain. The long cytoplasmic domain encodes two ITIMs (immunoreceptor tyrosine-based inhibitory motif) that bind SHP-1 and SHP-2 when phosphorylated and convey inhibitory activities to CEACAM1-4L. The short and long cytoplasmic domains have been shown to interact with intracellular molecules such as actin and calmodulin. Interestingly, CEACAM1 is expressed in a cell type and activation stage dependent manner with significant differences in the ratios of its short and long isoforms.

**Figure 2 cancers-11-00356-f002:**
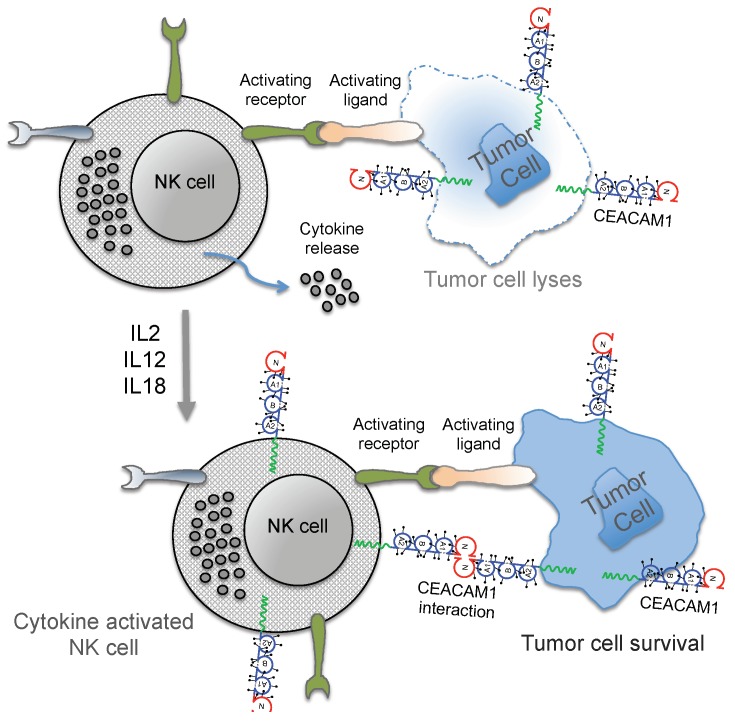
Summary of the natural killer (NK) activation mechanisms modulated by CEACAM1. (top) The interaction of activating receptors on NK cells with activating ligands present on malignant cells facilitates the cytotoxic activity of NK cells and, consequently, the tumor cell lyses. (down) CEACAM1 expression induced in cytokine activated NK cells interacts with tumor cell associated CEACAM1, thus inhibiting the cytolytic function of NK cells.

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
