# Peer review of "Size Matters: The Functional Role of the CEACAM1 Isoform Signature and Its Impact for NK Cell-Mediated Killing in Melanoma"

_cancers, 2019, doi:10.3390/cancers11030356_

Reviewer 1 Report

In this Review the authors describe the multifaceted role of CEACAM1 variants in the tumor progression and in the host immune response to tumors, with particular attention to the interactions occurring between NK cells and melanoma. The article is up-to-date and covers the major issues in the field; however the text is not well organized and sometime confusing. An extensive editing for English language and style is required.

Comments:

1) A figure describing structure and function of the different CEACAM1 isoforms, and a table that outlines their most relevant effects on the tumor and on NK cells may help to get clearer picture of what is described in the article.

2) The text needs a general revision. Several sentences should be edited or shortened for clarity, or revised to get appropriate connection with the context. See for example sentences starting at:

- line 32

- line 33

- line 49

- line 51

- line 66

- line 98

- line 105. In particular, this sentence indicates that CEACAM1 would induce NKG2D-L down-regulation. Actually, according to the Authors’ Cancer Research paper, different CEACAM1 isoforms can either inhibit or increase NKG2D-L expression on melanoma cells.

- line 111

- line 125

- line 136

- line 145

- line 187

- line 194

- line 228

- line 229

- line 233: the sentence is out of the context

- line 238 to line 257: all the sentences should be edited

- lines 270-276: by these sentences it is unclear whether CEACAM1 could be selectively induced on CD16neg NK cells or whether specific cytokines can induce CEACAM1 expression on different NK cell subsets.

- line 292

- line 294: this sentence is wrong. CEACAM1 inhibits cytolysis.

- line 298

- line 304 This sentence is not appropriate as the two papers (one is presently lacking – see the point 5 below) describe the role of CEACAM1 as additional mechanism to prevent auto-reactivity of NK cells in patients expressing low levels HLA class I molecules (TAP-deficient patients) or in conditions (maternal-fetal interface) requiring a strict control of immune effector functions.

3) Some terms should be modified or corrected:

- line 15: “leads” instead of “courses”

- line 36: “type” instead of “types”

- line 43: “gene copy number” instead of “copy number”

- line 111: “affinity” instead of “affine”

- line 217: “cycle” instead of “cycling”

- line 225: “functionally” instead of “functional”

- line 234: “endothelium” instead of “endothelial”

- line 239: “resolving”?

4) The adverb “however” has been inappropriately used in some sentences: lines 55, 103,

5) References.

- Ref. 13 does not correspond to what is stated in lines 64-66. That NK cells could kill melanoma targets expressing lower HLA class I molecules and higher activating NK receptor-Ligands was initially shown in: “Pende D. et al. Eur J. Immunol. 1998”; “Pende D. et al. Can. Res. 2002”. Ref. 13 should be moved to line 63.

- line 77: NCR and not NKC. The acronym NCR was introduced for the first time in “Moretta A. et al. Annu. Rev. Immunol. 2001”.

- Line 82: perhaps the authors refer to a paper by “Ali TH et al. Nature Comm. 2014”.

- line 292: The role of NKG2D in autoimmune diseases is treated in: “Guerra N. et al. Clin Immunol 2013”

- line 304: The role of CEACAM1 in pregnancy has been indicated in: “Markel G. et al. J Clin Invest 2002”

- lines 307-310: this sentence lacks the reference.

6) line 71: which cytokines? (only chemokines are indicated).

7) the section treating CEACAM1 structure and function is rather confusing. In particular the role of CEACAM1 and its variants in favoring or inhibiting tumorigenesis or tumor progression should be more systematically described.

Author Response

We sincerely appreciate the overall positive assessment of our work. A concerted effort was made to fully and adequately address all of the reviewers’ concerns, comments and suggestions. Following is a point-by-point verbatim response to the reviewers’ comments.

Reviewer #1:

In this Review the authors describe the multifaceted role of CEACAM1 variants in the tumor progression and in the host immune response to tumors, with particular attention to the interactions occurring between NK cells and melanoma. The article is up-to-date and covers the major issues in the field; however the text is not well organized and sometime confusing. An extensive editing for English language and style is required.

Main comment 1:

A figure describing structure and function of the different CEACAM1 isoforms, and a table that outlines their most relevant effects on the tumor and on NK cells may help to get clearer picture of what is described in the article.

Authors’ response:

We thank the Reviewer for giving the suggestion to include a Figure for clarifying the structure of the individual CEACAM1 isoforms. We have followed the recommendation and have generated a new Figure 1.

Main comment 2:

The text needs a general revision. Several sentences should be edited or shortened for clarity, or revised to get appropriate connection with the context. See for example sentences starting at:

- line 32

- line 33

- line 49

- line 51

- line 66

- line 98

- line 105.

In particular, this sentence indicates that CEACAM1 would induce NKG2D-L down-regulation. Actually, according to the Authors’ Cancer Research paper, different CEACAM1 isoforms can either inhibit or increase NKG2D-L expression on melanoma cells.

- line 111

- line 125

- line 136

- line 145

- line 187

- line 194

- line 228

- line 229

- line 233: the sentence is out of the context

- line 238 to line 257: all the sentences should be edited

- lines 270-276: by these sentences it is unclear whether CEACAM1 could be selectively induced on CD16neg NK cells or whether specific cytokines can induce CEACAM1 expression on different NK cell subsets.

- line 292

- line 294: this sentence is wrong. CEACAM1 inhibits cytolysis.

- line 298

- line 304 This sentence is not appropriate as the two papers (one is presently lacking – see the point 5 below) describe the role of CEACAM1 as additional mechanism to prevent auto-reactivity of NK cells in patients expressing low levels HLA class I molecules (TAP-deficient patients) or in conditions (maternal-fetal interface) requiring a strict control of immune effector functions.

Authors’ response:

With regard to the reviewer’s comment concerning the language, the revised manuscript version was has also been corrected by a native English speaker, Dr. Anthony Squire (added in the Acknowledgement section). We apologize for the confusing sentences and have therefore reworded  all of the parts indicated. For example, we rephrased the sentences in line 294 and 304, since we realized that our statement was misunderstood by the reviewer. We have also included the suggested references.

Main comment 3:

Some terms should be modified or corrected:

- line 15: “leads” instead of “courses”

- line 36: “type” instead of “types”

- line 43: “gene copy number” instead of “copy number”

- line 111: “affinity” instead of “affine”

- line 217: “cycle” instead of “cycling”

- line 225: “functionally” instead of “functional”

- line 234: “endothelium” instead of “endothelial”

- line 239: “resolving”?

Authors’ response:

We thank the Reviewer for the corrections and have adapted our manuscript accordingly.

Main comment 4:

The adverb “however” has been inappropriately used in some sentences: lines 55, 103

Authors’ response:

We have reworded our sentences at the lines indicated accordingly.

Main comment 5:

References.

- Ref. 13 does not correspond to what is stated in lines 64-66. That NK cells could kill melanoma targets expressing lower HLA class I molecules and higher activating NK receptor-Ligands was initially shown in: “Pende D. et al. Eur J. Immunol. 1998”; “Pende D. et al. Can. Res. 2002”. Ref. 13 should be moved to line 63.

- line 77: NCR and not NKC. The acronym NCR was introduced for the first time in “Moretta A. et al. Annu. Rev. Immunol. 2001”.

- Line 82: perhaps the authors refer to a paper by “Ali TH et al. Nature Comm. 2014”.

- line 292: The role of NKG2D in autoimmune diseases is treated in: “Guerra N. et al. Clin Immunol 2013”

- line 304: The role of CEACAM1 in pregnancy has been indicated in: “Markel G. et al. J Clin Invest 2002”

- lines 307-310: this sentence lacks the reference.

Authors’ response:

We gratefully thank the Reviewer for the suggested  references. We have replaced Ref. 13 by Pende et al., and have included the suggested references to our manuscript. The following references have also been added for the text between lines  307-310:

-Ullrich N, Heinemann A, Nilewski E, Scheffrahn I, Klode J, Scherag A et al. CEACAM1-3S Drives Melanoma Cells into NK Cell-Mediated Cytolysis and Enhances Patient Survival. Cancer Res 2015; 75(9):1897-1907.

-Ebrahimnejad A, Streichert T, Nollau P, Horst AK, Wagener C, Bamberger AM et al. CEACAM1 enhances invasion and migration of melanocytic and melanoma cells. Am J Pathol 2004; 165(5):1781-1787.

- Loffek S, Ullrich N, Gorgens A, Murke F, Eilebrecht M, Menne C et al. CEACAM1-4L Promotes Anchorage-Independent Growth in Melanoma. Front Oncol 2015; 5:234.

- Chen Z, Chen L, Baker K, Olszak T, Zeissig S, Huang YH et al. CEACAM1 dampens antitumor immunity by down-regulating NKG2D ligand expression on tumor cells. J Exp Med 2011; 208(13):2633-2640.

Main comment 6:

line 71: which cytokines? (only chemokines are indicated).

Authors’ response:

We thank the reviewer for this observation.Some examples of cytokines, which are involved during this process have now been added.

Main comment 7:

the section treating CEACAM1 structure and function is rather confusing. In particular the role of CEACAM1 and its variants in favoring or inhibiting tumorigenesis or tumor progression should be more systematically described.

Authors’ response:

We apologise for the confusing text concerning “CEACAM1 signaling and its function in melanoma” and have restructured this section accordingly.

In summary, we believe that we have now fully addressed all of the reviewers´ comments and suggestions, which have been most helpful in further improving the manuscript. We therefore look forward to the editors´ and reviewers´ comments concerning the revisions manuscript and are hopeful for   its final acceptance.

Thank you very much for your consideration.

Reviewer 2 Report

The authors have written an original and interesting review on the regulation of immunosurveillance by Natural Killer cells of melanoma and other tumors by the CEACAM1.

The review is dense, well documented and well organized in three parts: immunosurveillance and NK signaling in melanoma, CEACAM1 signaling and its function in melanoma and Regulation of NK cell function by CEACAM

The first part presents the particular immunogenicity of the melanoma and the new immunotherapies that have been developed to control /induce cytotoxic T cell responses. This part then presents the role of the Natural Killer cells and the advantages as cytotoxic anti-melanoma effectors. They describe the main NK receptors involved in the anti-melanoma function of NK cells.

The second part is longer and more detailed describes the complex signaling of the CEACAM1 molecule. The part is well written and documented. However, in its present form the presentation is very dense: subparagraphs and a diagram to recapitulate the main messages of the part should be included to improve the intelligibility of the signaling of the CEACAM1 molecule and its different isoforms

The part 3 clearly presents the conclusions and hypothesis of the authors on the involvement of CEACAM1 in NK immunosurveillance supported by convincing published results.

 Some references on metastatic lymph nodes infiltrating NK cells  should be added (Messaoudene M, 2014), as well as references on the effect of BRAF inhibition on NKG2D ligands by melanoma cells (Frazao, 2017, López-Cobo S, 2017)

Some sentences are very long lines 45-55, for example. In the sentence beginning line 50, what they (they are able to induce…) refers to?

What mean courses death in abstract ?

In part 1, line 86-89, human melanoma cell downregulate the NK cell receptors… the word ligands is required?

Author Response

We sincerely appreciate the overall positive assessment of our work. A concerted effort was made to fully and adequately address all of the reviewers’ concerns, comments and suggestions. Following is a point-by-point verbatim response to the reviewers’ comments.

Reviewer #2:

The authors have written an original and interesting review on the regulation of immunosurveillance by Natural Killer cells of melanoma and other tumors by the CEACAM1.

The review is dense, well documented and well organized in three parts: immunosurveillance and NK signaling in melanoma, CEACAM1 signaling and its function in melanoma and Regulation of NK cell function by CEACAM

The first part presents the particular immunogenicity of the melanoma and the new immunotherapies that have been developed to control /induce cytotoxic T cell responses. This part then presents the role of the Natural Killer cells and the advantages as cytotoxic anti-melanoma effectors. They describe the main NK receptors involved in the anti-melanoma function of NK cells.

The second part is longer and more detailed describes the complex signaling of the CEACAM1 molecule. The part is well written and documented. However, in its present form the presentation is very dense: subparagraphs and a diagram to recapitulate the main messages of the part should be included to improve the intelligibility of the signaling of the CEACAM1 molecule and its different isoforms

The part 3 clearly presents the conclusions and hypothesis of the authors on the involvement of CEACAM1 in NK immunosurveillance supported by convincing published results.

Main comment 1:

Some references on metastatic lymph nodes infiltrating NK cells  should be added (Messaoudene M, 2014), as well as references on the effect of BRAF inhibition on NKG2D ligands by melanoma cells (Frazao, 2017, López-Cobo S, 2017)

Authors’ response:

We thank the reviewer for the suggestion to include more information regarding NK cell impact in metastatic lymph nodes and the effect of therapeutic BRAF inhibition. Although we addressed this point from line 81 to line 91 in the original manuscript, which also included the suggested reference López-Cobo S, 2017 (former reference 21), we have now emphasized this point further by including additional information and references.

 Main comment 2:

Some sentences are very long lines 45-55, for example. In the sentence beginning line 50, what they (they are able to induce…) refers to?

Authors’ response:

We apologize for the long confusing sentences and have reworded part 45-55 accordingly. In addition, we have also included two new references (Kirkin AF et al., Exp Clin Immunogenet 1998; 15(1):19-32 and Passarelli A et al., Oncotarget 2017; 8(62):106132-106142) for the section referring to the induction of cytotoxic T cell-mediated immune response by melanoma cells.

Main comment 3:

What mean courses death in abstract?

Authors’ response:

We apologize for the typing mistake. “Courses” should be replaced by “causes”. Nevertheless, we have reworded this sentence also so it now reads, “It is characterized by continuously rising incidence and high mortality rate due to its high metastatic potential” (line 16-17).

Main comment 4:

In part 1, line 86-89, human melanoma cell downregulate the NK cell receptors… the word ligands is required?

Authors’ response:

We apologize for this mistake. We have reworded the section indicated so it now reads, “In consequence, recent in vitro data show that under pressure of the BRAF inhibitor Vemurafenib (PLX4032), human melanoma cells downregulate B7-H6, MICA, ULBP2 and the DNAM-1 ligand CD155, beside upregulation of MHC class I expression, to escape NK-cell mediated tumor cell recognition”.

In summary, we believe that we have now fully addressed all of the reviewers´ comments and suggestions, which have been most helpful in further improving the manuscript. We therefore look forward to the editors´ and reviewers´ comments concerning the revisions manuscript and are hopeful for   its final acceptance.

Thank you very much for your consideration.

Round  2

Reviewer 1 Report

The revised version of the manuscript has been improved in certain sections but still remains, as a whole, unsuitable for publication. In particular, section 1 is largely unsatisfactory as the text is often poorly organized and contains various mistakes especially on the biology of NK cells. Section 3 also needs further important actions.

Here are reported the most important points:

1) To my knowledge melanoma lesions, and not dysplastic nevi, can enter the vertical growth phase thus becoming pro-metastatic.

2) Lines 41-52. Although the general concepts on the immune system-tumor interplay can be argued by the text, this part remains hard to follow and largely incomplete. The description of the mechanisms of tumor cell transformation are interrupted by a sentence dealing with statistical data on metastatic patient survival; then a new sentence (introduced by the adverb “however”?!) describes the tumors as multicellular organs without any further detail on the tumor microenvironment and its unique cell types (which actually include different immune cells, generally polarized towards an immunoregulatory function, but also altered stromal cells).

3) Lines 53-57. As they stand these sentences are inconsistent: the immunotherapeutic strategies targeting PD1-PDL-1 “have revolutionized the field of melanoma therapy in recent years [10;11]. However, melanoma can evade the immune system” through the expression of PD-L1, which binds to PD1…”

4) As it stands, the sentence starting at line 62 is inconsistent and contains redundancies.

5) Lines 68-69. The ligands for activating receptors (and not the receptors!) are absent on healthy cells. Actually, ligands for activating NK-receptors may be expressed by certain cell types, such as myeloid cells or activated T cell blasts.

The HLA-I molecules, which are the ligands for major inhibitory NK receptors, are typically expressed by normal cells and ensure “tolerance towards a healthy self”.

By the way, the HLA-specific inhibitory receptors are never described in the manuscript; although, in some cases the inhibitory effects of CEACAM1 molecules could act in place of the HLA-KIR/NKG2A interaction (as in HLAneg tumor cells or in TAP-/- patients).

6) Lines 83-86. The inflammatory chemokines are not described. The authors should know that NK cells do not kill tumor cells via secretion of IL5, IL13, IL10. Granzymes and perforins are released in the context of the immunological synapsis, enter the target cell in contact with the NK cell and induce the proapoptotic cascade. IFNg and TNFa are released in the local microenvironment and mediate various effects on tumor cells, including, in some cases, cell death in surrounding tumor cells.

7) Lines 86-88. The sentence is misleading. It should be explained that Lymph Nodes generally include CD56brightCD16dim/neg poorly cytotoxic NK cells, while, in this case, CD56dimCD16+ highly cytotoxic NK cells have been activated and have increased the expression of CD56.

8) Line 108. In this sentence “However” makes no sense to me.

9) Lines 124-126. 1 IgV domain and 2 or 3 C domains for a total of 3 or 4 extracellular domains.

10) Lines 128-140. The description of the functional peculiarities of CEACAM1-L and -S isoforms is not well organized and contains redundancies (see for example lines 134-135: “…further supporting…”; and lines 137-138: “…further underlines…”. It is unclear which functions are supposed to be mediated by the -S isoforms. Which are the different functional effects of the different CEACAM1- L/L, L/S, S/S aggregations?

11) Line 141. CEACAM1-L (and not CEACAM1-S?) interacts with….

12) Lines 195-212. The authors describe the expression levels of CEACAM1 variants in melanoma and make contradictory conclusions on their role in the progression of the disease. The authors should explain/comment how to reconcile in vitro data, which indicate a predominant role for CEACAM1-L isoforms (line 201), and ex vivo analyses, which, instead, suggest a predominance of CEACAM1-S isoforms (line 212).

13) Lines 287-295. This part remains rather confusing, with sentences often organized without logical sequence. For example, the description of the effects of CEACAM1 on NKG2D-ligand expression is interrupted by a sentence on the general role of CEACAM1 as promising therapeutic target, then the plot moves to the effects of CEACAM1-CEACAM1 interactions on immune cells (which types? NK cells?) and finally goes back to the effects on NKG2D-ligands. Also, the sentence regarding the expression of CEACAM1 on CD16- NK cells is poor incisive: the authors should indeed explain the importance of the suppressive effect of CEACAM1 on the least cytotoxic NK cell subset.

Author Response

Dear Mr. Wang,

please find enclosed our second revised version of our manuscript entitled “Size matters: The functional role of the CEACAM1 isoform signature and its impact for NK cell-mediated killing in melanoma.

We sincerely appreciate the overall positive assessment of our work. A concerted effort has now been made to fully and adequately address the new aspects of Reviewer#1, the comments and suggestions. Following is a point-by-point verbatim response to the individual comments.

All changes have been marked in red to ensure recapitulation of our current version.

Regarding the comment that our manuscript should undergo extensive English editing, we would like to point out that the revised manuscript version has already been edited by Dr. Anthony Squire (Senior Researcher at the University Hospital Essen), who is an English native speaker.

We thank the second reviewer for his encouraging comments and address below the criticism of the first reviewer as follows:

Reviewer #1

Comments and Suggestions for Authors

The revised version of the manuscript has been improved in certain sections but still remains, as a whole, unsuitable for publication. In particular, section 1 is largely unsatisfactory as the text is often poorly organized and contains various mistakes especially on the biology of NK cells. Section 3 also needs further important actions.

Here are reported the most important points:

Comment 1:

To my knowledge melanoma lesions, and not dysplastic nevi, can enter the vertical growth phase thus becoming pro-metastatic.

Authors’ response:

Regarding the reviewer’s comment, we have realized that our introduction into the development and progression of metastatic melanoma could confuse the broad range of readership. Dysplastic nevi are classified as potential precursor lesions of melanoma, although they are nonobligate precursors, as based on histological observation, only a subset of melanomas have been found to arise out from dysplastic nevi. Therefore we rephrased the sentences in line 33-36 to ensure the misunderstanding of vertical growth phase induction by dysplastic nevi. We reworded the text as follows to be more precise:

“Melanomas often originate from benign nevi consisting of clonally expanded, highly proliferative melanocytes without the tendency for progression. The abnormal proliferation of melanocytes can lead to the development of dysplastic nevi, potential precursor lesions of melanoma, which are characterized by a low invasive potential. Early melanomas have the potential to metastasize to distant organs via induction of the vertical growth phase.”

 Comment 2:

Lines 41-52. Although the general concepts on the immune system-tumor interplay can be argued by the text, this part remains hard to follow and largely incomplete. The description of the mechanisms of tumor cell transformation are interrupted by a sentence dealing with statistical data on metastatic patient survival; then a new sentence (introduced by the adverb “however”?!) describes the tumors as multicellular organs without any further detail on the tumor microenvironment and its unique cell types (which actually include different immune cells, generally polarized towards an immunoregulatory function, but also altered stromal cells).

Authors’ response:

We thank the reviewer for this comment, rearranged individual parts to ensure a better understanding of the concept and added more cellular information about the composition of the microenvironment.

 Comment 3:

Lines 53-57. As they stand these sentences are inconsistent: the immunotherapeutic strategies targeting PD1-PDL-1 “have revolutionized the field of melanoma therapy in recent years [10;11]. However, melanoma can evade the immune system” through the expression of PD-L1, which binds to PD1…”

Authors’ response:

Although cancer immunotherapy has revolutionized the field of melanoma therapy, the response rate to the so called checkpoint blockers is still limited and around 40% in melanoma patients. Therefore, to our understanding, the implicated sentences are not inconsistent but should indicate that we still have the clinical problem that patients do not respond although they show high PD-L1 expression on the surface of melanoma cells. On the other hand, patients with very low or absent expression of PD-L1 on melanoma cells do respond to anti-PD-1 therapy. The underlying mechanism is still not understood and is therefore currently under investigation.

We altered the sentence into:

“PD-L1 interaction with PD1 causes immune tolerance through apoptosis of the activated lymphocytes. In consequence, high level of PD-L1 expression presented on the membrane of tumor cells has been correlated with poor prognosis in melanoma patients. Regarding therapeutic intervention by using checkpoint inhibitors, high PD-L1 expression on melanoma cells has been correlated with favorable clinical outcome. Nevertheless, recent data indicate that melanoma patients who lack expression of PD-L1 on melanoma cells can also respond to PD-1/PD-L1 inhibitors, although to a lower level”

Comment 4:

 As it stands, the sentence starting at line 62 is inconsistent and contains redundancies.

Authors’ response:

We rephrased this part to take into account the reviewer’s comment.

Comment 5:

Lines 68-69. The ligands for activating receptors (and not the receptors!) are absent on healthy cells. Actually, ligands for activating NK-receptors may be expressed by certain cell types, such as myeloid cells or activated T cell blasts.

The HLA-I molecules, which are the ligands for major inhibitory NK receptors, are typically expressed by normal cells and ensure “tolerance towards a healthy self”.

By the way, the HLA-specific inhibitory receptors are never described in the manuscript; although, in some cases the inhibitory effects of CEACAM1 molecules could act in place of the HLA-KIR/NKG2A interaction (as in HLAneg tumor cells or in TAP-/- patients).

Authors’ response:

We have addressed the first two points in the revised version of our manuscript but skipped the third part due to manuscript length limitations. In addition, this point seems still to be too speculative.

Comment 6:

Lines 83-86. The inflammatory chemokines are not described. The authors should know that NK cells do not kill tumor cells via secretion of IL5, IL13, IL10. Granzymes and perforins are released in the context of the immunological synapsis, enter the target cell in contact with the NK cell and induce the proapoptotic cascade. IFNg and TNFa are released in the local microenvironment and mediate various effects on tumor cells, including, in some cases, cell death in surrounding tumor cells.

Authors’ response:

We thank the reviewer for his corrections. Obviously we have been too inaccurate because we wanted to keep this part as short as possible. We have now picked up the reviewer’s suggestion and have changed our text as follows:

“Furthermore, NK cell accumulation in tumor-infiltrated lymph nodes of melanoma patients results in NK cell-mediated killing via the release of perforin and granzyme. Perforin and granzyme enter the target cell in contact with the NK cell and induce apoptosis. Additionally, chemokines such as CCL3/MIP-1α and CCL4/MIP-1β, and pro-inflammatory cytokines such as TNF-α, IFN-γ, IL-5, IL-10 and IL-13 are secreted in the local microenvironment and mediate various effects including recruitment of inflammatory cells to the tumor and in some cases even cell death in surrounding tumor cells”.

Comment 7:

Lines 86-88. The sentence is misleading. It should be explained that Lymph Nodes generally include CD56brightCD16dim/neg poorly cytotoxic NK cells, while, in this case, CD56dimCD16+ highly cytotoxic NK cells have been activated and have increased the expression of CD56.

Authors’ response:

Regarding the reviewer’s comment, we have rephrased this sentence to make the statement more precise. We have changed the sentence as follows:

“Additionally, the detection of a specific CD56brightCD16+NK cell subset in metastatic lymph nodes which generally include CD56brightCD16dim/neg poorly cytotoxic NK cells suggested that nodal NK cells may represent an attractive therapeutic target for advanced melanoma patients”.

Comment 8:

Line 108. In this sentence “However” makes no sense to me.

Authors’ response:

We removed this “however” in the revised version.

Comment 9:

Lines 124-126. 1 IgV domain and 2 or 3 C domains for a total of 3 or 4 extracellular domains.

Authors’ response:

We have altered the text as follows: “All CEACAM1 variants share one membrane distal IgV-like domain (N-domain) modulating homophilic or heterophilic interactions, and two or three IgC-like domains for a total of 3 (CEACAM1-3) or 4 (CEACAM1-4) heavily glycosylated extracellular domains”.

Comment 10:

Lines 128-140. The description of the functional peculiarities of CEACAM1-L and -S isoforms is not well organized and contains redundancies (see for example lines 134-135: “…further supporting…”; and lines 137-138: “…further underlines…”. It is unclear which functions are supposed to be mediated by the -S isoforms. Which are the different functional effects of the different CEACAM1- L/L, L/S, S/S aggregations?

Authors’ response:

As kindly suggested by the reviewer, we have reorganized and expanded the text as follows to be more precise. We also transferred the section to be more chronological. 

“All these findings underline the different functional potential of CEACAM1-L and –S. What makes the CEACAM1 driven signaling even more complex is the fact that CEACAM1 dimerizes not only in ‘trans’ but also in ‘cis’, giving rise to L/L, L/S and S/S constellations, which seem to cause different functional effects. Öbrinks group very nicely showed that the trans-homophilic binding mediated by the N-domain induces cis-dimerization of CEACAM1. The cis-homodimerization of CEACAM1-L brings its cytoplasmic domains together thus changing the relative binding affinities for SHP-1/2 and c-Src. Consequently, increased CEACAM1-L homodimerization results in increased binding and activation of SHP-1/2. CEACAM1-S can interfere with the CEACAM1-L homodimer by forming a heterodimer with CEACAM1-L thus decreasing the level of the CEACAM1-L cytoplasmic dimer configuration. Consequently the CEACAM1 driven signaling mechanisms and functional outcome can be modulated by variations of the CEACAM1-S/CEACAM1-L expression ratios”.

Comment 11:

Line 141. CEACAM1-L (and not CEACAM1-S?) interacts with….

Authors’ response:

We have altered the text as follows to reflect the facts more clearly:

“Important to note, CEACAM1-L, but not CEACAM1-S, negatively regulates proliferation via its ITIM domain. It also binds paxillin if tyrosine phosphorylated and contains a talin and β-catenin binding site in its cytoplasmic tail. A common feature of CEACAM1-L and -S is that they interact with the actin cytoskeleton and further proteins like, tropomyosin, filamin A, and calmodulin”.

Comment 12:

Lines 195-212. The authors describe the expression levels of CEACAM1 variants in melanoma and make contradictory conclusions on their role in the progression of the disease. The authors should explain/comment how to reconcile in vitro data, which indicate a predominant role for CEACAM1-L isoforms (line 201), and ex vivo analyses, which, instead, suggest a predominance of CEACAM1-S isoforms (line 212).

Authors’ response:

Regarding the reviewer’s comment, we have the impression that our data were misunderstood by the reviewer. First, we described that we analyzed cell lines which were generated from melanoma tissue. Here, we found that all CEACAM1-positive cell lines expressed the CEACAM1-4L variant alone or in combination with other CEACAM1 variants. Moreover, the expression levels of the L-variants (both CEACAM1-3L and CEACAM1-4L) dominated against the S-variants. Via genetically introduced CEACAM1-4L expression into originally CEACAM1 negative melanoma cell lines, we observed increased migratory and invasive growth potential, supporting the role of CEACAM1-4L in melanoma progression and metastasis (please see line 251-260).

Regarding our analyses of tissue biopsies, CEACAM1-4L and CEACAM1-4S expression could be already detected in early tumor development, implicating, that these isoforms could have the potential to drive melanoma cell progression from early time on, which would fit with our in vivo observations of the genetically modified cell lines (CEACAM1-4L introduction drives anchorage-independent growth and migratory/invasive potential). At later stages of melanoma progression, the expression of CEACAM1-3L and CEACAM1-3S is also detectable BUT ONLY in combination with the already existing isoforms CEACAM1-4L and CECAM1-4S. Thus, it is not the case that expression of CEACAM1-4L and CEACAM1-4S is only visible in early melanoma. The detailed mechanism underlying the induction and expression of the individual CEACAM1 variants in melanoma is largely unknown and currently under investigation in our lab.

Comment 13:

Lines 287-295. This part remains rather confusing, with sentences often organized without logical sequence. For example, the description of the effects of CEACAM1 on NKG2D-ligand expression is interrupted by a sentence on the general role of CEACAM1 as promising therapeutic target, then the plot moves to the effects of CEACAM1-CEACAM1 interactions on immune cells (which types? NK cells?) and finally goes back to the effects on NKG2D-ligands. Also, the sentence regarding the expression of CEACAM1 on CD16- NK cells is poor incisive: the authors should indeed explain the importance of the suppressive effect of CEACAM1 on the least cytotoxic NK cell subset.

Authors’ response:

We apologize for the confusion and have rearranged this part for a better understanding.

Thank you very much for your consideration.

Sincerely yours,

Iris Helfrich Ph.D.

Round  3

Reviewer 1 Report

The manuscript has been significantly modified by the authors who replied to most of the raised questions.

Only two additional minor points:

1) The sentence starting at line 301 should be moved (and adapted) to line 313: "...to functional inhibition of those cells (131-134). Following activation it is mainly ..."

"...Additionally, evidence has amassed that CEACAM1 is involved in the regulation of the NKG2D-ligand expression (141-143). Indeed, CEACAM1 can down-regulate MICA/B and ULBP1 from the tumor cell surface thus influencing their immunogenicity (135, 136). Several ..."

2) Pay attention to reference list, which has completely lost the setting.

Author Response

Neal Wang

Assistant Editor

- through the online submission system -

 Third Minor Revision of the Manuscript ID: cancers-410201

Dear Mr. Wang,

please find enclosed the third revised version of our manuscript entitled “Size matters: The functional role of the CEACAM1 isoform signature and its impact for NK cell-mediated killing in melanoma.

We sincerely appreciate the implicated minor points and suggestions of Reviewer#1. Following the point-by-point response is listed to the indicated comments.

All changes have been marked in red to ensure tracking of our current version.

Reviewer #1

The manuscript has been significantly modified by the authors who replied to most of the raised questions.

Only two additional minor points:

1) The sentence starting at line 301 should be moved (and adapted) to line 313: "...to functional inhibition of those cells (131-134). Following activation it is mainly ..."

"...Additionally, evidence has amassed that CEACAM1 is involved in the regulation of the NKG2D-ligand expression (141-143). Indeed, CEACAM1 can down-regulate MICA/B and ULBP1 from the tumor cell surface thus influencing their immunogenicity (135, 136). Several ..."

2) Pay attention to reference list, which has completely lost the setting.

Authors’ response:

1.) Regarding the reviewer’s suggestion, we rearrange the indicated text as follows (line 301-313):” Following activation it is mainly the CD16-negative NK subset that expresses CEACAM1 and cells expressing the CEACAM1 are protected from lysis by CEACAM1-positive NK cells [101;105]. Important to note, while freshly isolated T- and B-cells express very low amounts of CEACAM1, freshly isolated NK cells were CEACAM1 negative [106]. However, the CEACAM1 expression on NK cells (and T cells) depends on their differentiation stage and can be strongly enhanced by activation with interleukin-2, (IL-2) IL-12 and IL-18, which may lead to distinct regulatory mechanisms of the functional subpopulations of NK cells (Figure 2) [107;108]. The central meaning of the homophilic CEACAM1 interaction in trans was also supported by reports showing that CEACAM1 positive NK cells inhibited cytolysis of CEACAM1 expressing tumor cells by impairing NKG2Ds ability to stimulate cytolysis. Additionally, evidence has amassed that CEACAM1 is involved in the regulation of the NKG2D-ligand expression [72;101;109]. Indeed, CEACAM1 can down-regulate MICA/B and ULBP1 from the tumor cell surface thus influencing their immunogenicity [110;111].”

2.) As already mentioned, caused by IT updates Word lost the connection to the ReferenceManager Software which resulted in temporary confusion of the listed References.

We have solved this problem and have now carefully corrected the implicated references and reference list.

We hope that the current version will be acceptable for publication and are looking forward to a positive decision.

Thank you very much for your consideration.

Sincerely yours,

Iris Helfrich Ph.D.